# Fungus-Based MnO/Porous Carbon Nanohybrid as Efficient Laccase Mimic for Oxygen Reduction Catalysis and Hydroquinone Detection

**DOI:** 10.3390/nano12091596

**Published:** 2022-05-08

**Authors:** Haoran Ge, Hailong Zhang

**Affiliations:** School of Electrical and Automation Engineering, Nanjing Normal University, Nanjing 210023, China; haogran@163.com

**Keywords:** fungus, nanozyme, laccase, oxygen reduction reaction, hydroquinone detection

## Abstract

Developing efficient laccase-mimicking nanozymes via a facile and sustainable strategy is intriguing in environmental sensing and fuel cells. In our work, a MnO/porous carbon (MnO/PC) nanohybrid based on fungus was synthesized via a facile carbonization route. The nanohybrid was found to possess excellent laccase-mimicking activity using 2,2′-azinobis (3-ethylbenzthiazoline-6-sulfonic acid) diammonium salt (ABTS) as the substrate. Compared with the natural laccase and reported nanozymes, the MnO/PC nanozyme had much lower *K*_m_ value. Furthermore, the electrochemical results show that the MnO/PC nanozyme had high electrocatalytic activity toward the oxygen reduction reaction (ORR) when it was modified on the electrode. The hybrid nanozyme could catalyze the four-electron ORR, similar to natural laccase. Moreover, hydroquinone (HQ) induced the reduction of oxABTS and caused the green color to fade, which provided colorimetric detection of HQ. A desirable linear relationship (0–50 μM) and detection limit (0.5 μM) were obtained. Our work opens a simple and sustainable avenue to develop a carbon–metal hybrid nanozyme in environment and energy applications.

## 1. Introduction

The oxygen reduction reaction (ORR) pathway is of great importance in meeting the specific needs of various ORR-related systems, such as biomimetic catalysis, fuel cells, and biosensing [1,2,3,4,5]. Using oxygen in aerobic metabolism under mild conditions is very attractive since O_2_ in the triplet ground state is very stable. Miraculously, metalloenzymes have provided an efficient and alternative solution to this obstacle under mild conditions [6,7,8]. As important metalloenzymes, laccases with copper as active sites catalyzed the oxidation of diverse biologicall and environmental substrates with oxygen [9,10,11]. In these reactions, laccase had the ability to catalyze the four-electron reduction of oxygen to water. Therefore, laccase has a wide application in the environmental and biosensing fields related to ORR [12,13,14]. More notably, laccase has attracted much attention in the research field of biofuel cells and bioelectrochemistry because it can realize the electrochemical catalytic reduction of oxygen molecules at low overpotential [15,16]. However, due to its fragile structure, the application of laccase in these fields is often limited by inherent defects, such as high cost, short active life, poor environmental tolerance, etc.

As new and robust artificial enzymes, nanozymes with highly specialized biological functions have wide practical applications, ranging from cancer therapy to biosensing, because of their advantages of catalytic activity and stability, cheap costs, and special physical–chemical properties [17,18,19]. The enzyme-like activity of nanozymes is flexibly related to its composition, structure, and size. Until now, some copper-based nanozymes have been reported to exhibit laccase-mimicking activity [20,21,22], but most nanozymes, such as gold nanoparticles, iron oxide nanoparticles, and carbon-based nanozymes, lack laccase-mimicking activity [23,24]. Therefore, developing laccase-mimicking nanozymes that are low-cost and have a simple operation process, good biocompatibility, high activity, and stability is intriguing to pursue but remains challenging.

Due to its stable physical–chemical properties and three-dimensional interconnected porous structure, porous carbon material provides effective channels for material transmission, liquid flow, and gas diffusion, making it an important functional material used in adsorption, separation, sensing, energy storage, and catalysis [25,26,27]. Considering environmental and sustainable development problems, it is both a huge challenge and encouraging to use biomass-based carbon materials as the primary materials for chemical and enzymatic conversion processes [28,29]. Because widely distributed fungi mainly consist of polysaccharides, protein, and amino acids, they have both high-content carbon and heteroatom elements, such as oxygen, nitrogen, and sulfur. Therefore, they are economical, environmentally friendly, and sustainable carbon precursors for preparing heteroatoms-enriched carbon materials with the ability to complex with metal ions.

Recently, the enzyme-mimicking activity of manganese oxides (MnO_x_) has received considerable interest due to their attractive properties of being rich, cheap, a green alternative, and active, but the laccase-mimicking activity of manganese oxides remains scarce [30,31,32]. Our work used a biological fungus with a rich source and low price as raw material to prepare the MnO/PC hybrid (Figure 1). The nanohybrid exhibited superior laccase-mimicking activity and high electrocatalytic activity for ORR. The combination of the respective properties of the two components achieved the enhanced catalytic performances of the hybrid nanozyme cooperatively. To the best of our knowledge, this is the first report to address a fungus-based nanozyme with laccase-mimicking activity, which provides a facile and sustainable strategy for expanding a carbon–metal hybrid nanozyme in environment and energy applications.

## 2. Experimental Section

### 2.1. Chemicals

Tremella produced in Fujian province was purchased from a local supermarket (Yangzhou, China). Pt/C (20 wt%) and 2,2′-azino-bis (3-ethylbenzothiazoline-6-sulfonic acid) (ABTS) were purchased from Sigma. Other reagents and chemicals were reagent-grade and were acquired from Sinopharm Chemical Reagent Co., Ltd. (Shanghai, China). Deionized water was used throughout.

### 2.2. Instrumentation and Characterization

X-ray diffraction (XRD) analysis was performed with a Bruker AXS D8 ADVANCE X-ray diffractometer (Germany). Scanning electron microscopy (SEM) images and energy-dispersive X-ray (EDX) were conducted on a Zeiss Supra 55 (Germany). X-ray photoelectron spectroscopy (XPS) measurements were conducted on an ESCALAB-MKII 250 photoelectron spectrometer (UK), using Al/Kα as the X-ray source for excitation. UV-vis spectra were recorded using a UV-2501 (Japan). The structure was analyzed by Raman spectroscopy (DXR, GX-PT-2412, Thermo, USA) with a 532 nm laser as the excitation wavelength.

### 2.3. Synthesis of MnO/PC Nanohybrid

As shown in Figure 2, the obtained tremella was first cleaned and soaked in deionized water for 2 h and was then heated for more than 30 min until it became sticky. A 0.5 M manganese acetate solution was mixed with the sticky tremella. The mixture was stirred for about half an hour at room temperature and left for 48 h. The mixed system was frozen at −90 °C for 48 h to be dehydrated with a freeze-dryer. Next, the as-prepared sample was carbonized at 800 °C for 2 h with a ramping rate of 5 °C min^−1^ in the protection of N_2_. The sample was then washed with ethanol and water and finally dried at 60 °C. A MnO/PC-700 nanohybrid was synthesized under the same procedure, except that 700 °C was used as the calcination temperature. The porous carbon was obtained via the same synthetic procedure, except for the absence of manganese ions.

### 2.4. Laccase-Mimicking Activity of MnO/PC Nanohybrid

The laccase-mimicking activity of the MnO/PC nanohybrid was measured using catalytic oxidation of the typical laccase substrate ABTS at 20 °C. For a typical oxidation experiment, 50 μL MnO/PC dispersion (5 mg/mL) and 60 μL 20 mM ABTS were introduced in 2.890 mL of 0.2 M HAc-NaAc at pH 4.0. The UV-vis spectra of the system were recorded after 20 min.

### 2.5. ORR Electrocatalytic Activity of MnO/PC Nanohybrid

Electrochemical measurements were recorded on an electrochemical workstation (VMP3, Biologic, Seyssinet-Pariset, France) with a conventional three-electrode system. The electrode system was composed of a Pt wire as the counter electrode, Ag/AgCl as the reference electrode, and a rotating ring-disk electrode (RRDE) as the working electrode. The detailed electrochemical procedures are shown in the Appendix A.

### 2.6. Determination of Hydroquinone

For a typical determination experiment, 50 μL MnO/PC dispersion (5 mg/mL) and 60 μL of 10 mM ABTS were introduced in 0.2 M HAc-NaAc at pH 4.0. After reaction for 10 min, ABTS turned into green oxABTS. Next, HQ of different concentrations was added, and the green color faded. Two minutes later, the concentration of HQ was obtained according to the absorbance at 416 nm. The lake real sample was filtered and introduced in 0.2 M HAc-NaAc at pH 4.0, and the same procedure was used to analyze the HQ concentration in the system.

## 3. Results and Discussion

### 3.1. Characterizations of MnO/PC Nanohybrid

As a typical fungus, tremella consisting mainly of polysaccharides was selected as the precursor and template for obtaining the carbon-loaded metal nanohybrid with high-temperature pyrolysis. Figure 2 summarizes the synthesis strategy of the formation of the hybrid. Tremella rich in oxygen-containing functional groups contributes to the specific affinity with Mn(II) ion and the formation of carbon loaded with the MnO hybrid after the calcination.

The microstructure of the MnO/PC nanohybrid, based on the fungus’s carbon obtained at 800 °C, was first characterized by scanning electron microscopy (SEM). As seen in Figure 1A,B, the MnO/PC nanohybrid exhibited macropores, and shell-like structured MnO crystals were loaded on the carbon surface. The SEM image with high magnification shows that the shell-like structured MnO was composed of many nanoparticles with small sizes. For the individual biomass carbon obtained in the absence of metal ion (Appendix A), only the porous carbon was observed, and the surface of the carbon was approximately smooth. The EDX result demonstrated that the nanohybrid was composed of Mn, C, and O elements (Figure 1C). Different from many earlier porous carbon nanomaterials based on biomass cellulose, the process of KOH activation was avoided when tremella was used as the biomass.

The wide-angle X-ray diffraction (XRD) pattern of the hybrid is shown in Figure 1D. The broad peak between 20 to 30° indicates the amorphous carbon. Five diffraction peaks at 34.91, 40.55, 58.72, 70.17, and 73.70° indexed to (111), (200), (220), (311), and (222) crystal planes of MnO (JCPDS no. 07-0230) indicate that MnO was formed in the system. In addition, since the relative intensity of the MnO peaks was weak, the shell-like structured MnO in the nanohybrid may have poor crystallinity. Because no other diffraction peaks were obtained, pure MnO was present. Raman spectroscopy was also used to characterize the structure of the nanohybrid (Appendix A). As seen, the characteristic peak at 646.8 cm^−1^ was attributed to the Mn–O vibration [33]. Since the two typical peaks centering at 1348.5 cm^−1^ (D-band) and 1590.6 cm^−1^ (G-band) with the relative intensity ratio of the D-band to the G-band (I_D_/I_G_) was 0.88, the graphitic carbon phase was present in the nanohybrid [34].

XPS spectra were used to demonstrate the surface composition and chemical valence state of the MnO/PC nanohybrid. The full-scan spectrum demonstrated that the hybrid was also mainly composed of the elements Mn, O, and C (Figure 1E), suggesting the formation of a manganese oxide and carbon hybrid. The surficial atomic concentration of the Mn, C, and O elements of the hybrid was about 6.79%, 76.16%, and 17.05%, respectively. According to the high-resolution Mn 2p XPS spectrum of the hybrid (Figure 1F), the peaks at 640.3 and 652.1 eV were caused by Mn 2p_1/2_ and Mn 2p_3/2_, respectively, suggesting the formation of MnO. Furthermore, an energy separation of 11.8 eV also demonstrated Mn 2p_3/2_ and Mn 2p_1/2_ in MnO [35,36]. The presence of MnO was further confirmed by the O 1s spectrum. As seen in Appendix A, three typical peaks at 530.1, 531.6, and 532.9 eV are caused by Mn-O-Mn, Mn-O-H, and H-O-H, respectively.

### 3.2. Laccase-Mimicking Activity of MnO/PC Nanohybrid

After characterizing its structure, the laccase-mimicking activity of the MnO/PC nanohybrid using the typical ABTS as the substrate was measured. Figure 2A shows that the MnO/PC nanohybrid could catalyze the oxidation of ABTS to form the typical green color rapidly. On the contrary, the individual ABTS system was nearly colorless. There was a significantly strong absorbance at 416 nm of the system with the hybrid nanozyme, indicating that the MnO/PC nanohybrid had superior laccase-mimicking activity. By comparison, the laccase-mimicking activity of PC and Mn^2+^ was studied and we found that PC could only weakly catalyze ABTS oxidation and that Mn^2+^ had no catalytic activity for ABTS oxidation under the same experimental conditions.

To confirm the role of the MnO/PC nanohybrid obtained at 800 °C in determining laccase-like activity, the catalytic activity of the MnO/PC-700 nanohybrid was further studied. Appendix A shows that the porous structure of carbon and the structure of MnO changed greatly at the calcination temperature of 700 °C. As shown in Appendix A, the MnO/PC-700 nanohybrid can also catalyze the oxidation of ABTS to produce green color, displaying absorption at 416 nm. Because the laccase-like activity of MnO/PC-800 was higher than that of MnO/PC-700, the cooperative effect of the porous-structured carbon and shell-like structured MnO was favorable for the substrates to bind and activate. Notably, the laccase-mimicking activity of the hybrid remained almost unchanged after more than one year, indicating the high stability of the hybrid nanozyme.

Similar to laccase, the enzyme-like activity of MnO/PC nanohybrid depended on the pH and temperature. As seen in Appendix A, the laccase-mimicking activity of the nanozyme decreased with the pH but still exhibited more than 90% activity at pH 5.0. As seen in Appendix A, the laccase-mimicking activity of MnO/PC nanohybrid increased with the increase of temperature. Because the laccase-mimicking activity of the hybrid did not change much in the temperature range of 20–35 °C, it was free to use the hybrid nanozyme at room temperature.

The kinetic parameters were obtained to investigate the laccase-mimicking activity of the MnO/PC nanohybrid. The typical Michaelis–Menten curves were obtained within certain concentration ranges of ABTS (Figure 2B), and then *K*_m_ and *V*_max_ were obtained according to the Lineweaver–Burk linear fitting equation (Table 1). As seen, the MnO/PC nanohybrid had a much lower *K*_m_ and higher *V*_max_ [37,38,39,40]. Since pure PC exhibited poor laccase-like activity, the presence of MnO provided active sites for the laccase-mimicking activity of the MnO/PC nanohybrid. Recently, the enzyme-like activity of manganese oxides (MnOx) has received considerable attraction due to its outstanding catalytic performance. Wang, et al., reported the laccase-like activity of manganese oxides such as MnO_2_ and Mn_3_O_4_ using ABTS as the substrate [40], but the laccase-like activity of MnO has never been reported. As seen in Table 1, the *K*_m_ and *V*_max_ for the laccase-like activity of the MnO/PC nanohybrid with ABTS as the substrate was much better than that of the reported manganese oxides. Thus, the synergistic effects of MnO and PC contributed to the superior catalytic activity of the MnO/PC hybrid. For the MnO/PC nanohybrid with the integration of PC and MnO, the porous structure in the catalysts provided a path for the reactant and product molecules to diffuse in and out freely and resulted in high affinity towards substrates [41,42,43]. Furthermore, the shell-like MnO distributed on the porous carbon surface was composed of numerous nanoparticles with small sizes. The dispersed hierarchical structure enabled plentiful active sites provided by small-sized MnO nanoparticles, accessible to the substrates during the reaction [44,45]. These factors all led to the high laccase-like activity of the hybrid nanozyme. Notably, the *K*_m_ for the laccase-like activity of the MnO/PC nanohybrid with ABTS as the substrate was similar to the *K*_m_ for the 3DGF/Pt NC nanohybrid. This further confirmed that the porous structure of carbon greatly contributed to the high affinity of the substrate toward the nanozymes [39]. Evidently, the fungus-based MnO/PC hybrid nanozyme had the advantages of being low cost, requiring a simple operation process, and being renewable. In addition to a laccase-mimicking nanozyme, the use of immobilized natural laccase is also important in practice. For comparison, the kinetic parameters of the immobilized laccase with ABTS as the substrate were provided. Table 1 shows that the MnO/PC nanohybrid had relatively higher affinity and catalytic activity than that of the immobilized natural laccase. Thus, as a kind of laccase-like mimic with high catalytic activity, the MnO/PC nanohybrid has potential application in practice. 

### 3.3. ORR Electrocatalytic Activity of MnO/PC Nanohybrid

Motivated by the superior laccase-like activity of the MnO/PC nanohybrid, the ORR electrocatalytic activity of the obtained laccase-mimicking MnO/PC nanohybrid was further studied with the electrochemical method. The ORR electrocatalytic activities of the MnO/PC nanohybrid were evaluated in O_2_-saturated 0.1 M KOH using a rotating ring-disk electrode (RRDE). The CV profiles for the MnO/PC nanohybrid in N_2_ and O_2_ saturated 0.1 M KOH are given in Figure 3A. An obvious peak was observed in the O_2_-saturated 0.1 M KOH solution, but no peak was obtained in the N_2_-saturated KOH solution. Thus, the MnO/PC nanohybrid exhibited ORR electrocatalytic activity.

The linear sweep voltammetry (LSV) curves of the porous carbon and MnO/PC nanohybrid are shown in Appendix A. As seen, the MnO/PC nanohybrid showed notably enhanced performances, i.e., increased limiting currents and half-wave potentials. Compared with individual porous carbon, the superior ORR activity of the MnO/PC nanohybrid may be ascribed to the facile mass transport of the reactants and electrons caused by the porous structure and synergistic effect between MnO and PC. Figure 3B presents the LSV curves of the MnO/PC nanohybrid with various rotating rates from 400 to 2025 rpm and the corresponding K–L plots of different potentials. It can be seen from a set of LSV curves of MnO/PC that the current density increased when the rotating speed increased (Figure 3B). The current density achieved the diffusion, limiting current well at different rotation speed. The Koutecky–Levich (K–L) plots under different potentials displayed excellent linearity, and there was almost no difference between these plots, suggesting first-order reaction kinetics with respect to the dissolved oxygen in electrolyte and similar electron transfer number *n* at different potentials of the hybrid nanozyme. The transferred electron number *n* for the MnO/PC hybrid nanozyme was analyzed to be ca. 4 at the potential range from 0.20 to 0.50 V, demonstrating that the ORR process of the MnO/PC nanohybrid followed a typical four-electron pathway.

The LSV curves of the MnO/PC nanohybrid, MnO/PC-700 nanohybrid, and 20 wt% Pt/C are shown in Figure 3C. The half-wave potential for the MnO/PC hybrid was about 0.73 V, with the limiting currents of −5.87 mA cm^−2^. The half-wave potential difference for the hybrid and 20 wt% Pt/C was only about 60 mV, and the limited current of MnO/PC nanohybrid was almost close to that of 20 wt% Pt/C. Compared with the MnO/PC nanohybrid, the MnO/PC-700 nanohybrid displayed a more negative half-wave potential and a lower limiting current density, which further revealed that the cooperative effect of MnO and PC in the nanohybrid was helpful to improve the ORR catalytic performance significantly. A comparison of the ORR electrochemical performances for the MnO/PC nanohybrid with the relevant catalysts and Pt/C is provided in Appendix A. As seen, the ORR electrochemical performances of the MnO/PC nanohybrid was much better than that of the reported pure MnO catalyst. Compared with the most reported MnO/C hybrid, the MnO/PC nanohybrid exhibited an efficient four-electron pathway and high limiting current density, close to the commercial Pt/C-20%.

According to the ring and disk current of the MnO/PC nanohybrid, MnO/PC-700 nanohybrid, and 20 wt% Pt/C (Figure 3C), the transferred electron number (n) and percentage yield of HO_2_^−^ are shown in Figure 3D. The electron number higher than 3.88 in the scanning potential range of 0.2–0.6 V was coincidental with the K–L plot results, further indicating a four-electron electrochemistry reaction process. Simultaneously, the percentage yield of HO_2_^−^ for the MnO/PC nanohybrid in alkaline environments was lower than 7.5%, which was close to that of the 20 wt% Pt/C. The MnO/PC-700 exhibited a lower *n* (3.62) and higher HO_2_^−^ yields (12.5–22.0%) in the scanning potential range of 0.2–0.6 V, demonstrating that the MnO/PC nanohybrid can provide a high ORR activity and an efficient four-electron pathway. This further confirmed that the structure of a MnO/PC nanohybrid matters greatly in determining the superior catalytic activity. As seen in Appendix A, the nanohybrid in the ORR electrochemical activity is durable. Because the MnO/PC nanohybrid catalyzed the four-electron ORR efficiently, similar to natural laccase, the nanozyme has potential as a cathode catalyst for fuel cells.

### 3.4. Sensing HQ Based on MnO/PC Nanohybrid

Since hydroquinone (HQ) has been recognized as one of the biggest environmental pollutants, it is important to fabricate a facile and sensitive method to detect HQ. Although nanozymes have been widely applied in sensing environmental pollutants with the colorimetric method, the laccase-mimicking nanozyme is seldom used to detect phenols. Due to the superior laccase-like activity of the MnO/PC nanohybrid, it is expected to fabricate HQ colorimetric sensors through regulating the catalytic activity of the nanohybrid.

The MnO/PC nanohybrid catalyzed the oxidation of ABTS into green-colored oxABTS effectively at room temperature, but the addition of HQ into the system led to a fading blue color, with a decrease in the absorbance at about 416 nm (Figure 4A). With the increase in HQ concentration, the decrease in the absorbance was enhanced. Based on this, the colorimetric detection for HQ concentration was developed using the absorbance decrease of ABTS. Figure 4B shows the calibration curve for HQ, and the absorbance at 416 nm decreased linearly with the increasing concentration of HQ in the range of 0 to 50 μM, with a detection limit of 0.2 μM (S/N = 3).

As seen in Table 2, HQ colorimetric sensors have been developed using different nanozymes. Although a HQ colorimetric sensor based on the peroxidase-like and oxidase-like activity of a nanozyme with 3,3′,5,5′-tetramethylbenzidine (TMB) as the substrate has been widely reported in some earlier studies, studies on the HQ colorimetric sensor based on laccase-like nanozymes with ABTS as the substrate are scarce. Compared with the earlier studies, the proposed calorimetric sensor exhibited a desirable LOD and linear range. Furthermore, since the hybrid nanozyme was synthesized via a facile and sustainable route, the colorimetric laccase mimetic strategy has potential applications in environmental fields.

The mechanism of sensing HQ using the chromogenic substrate ABTS and the MnO/PC nanohybrid-catalyzed ABTS color reaction is shown in Figure 3. First, the color of the ABTS changed into green-colored oxABTS due to the superior laccase-like activity of the MnO/PC nanohybrid, which was accompanied by a sharp increased absorbance at 416 nm (Figure 3a). Second, due to the strong reducibility of HQ, the redox reaction occurred between HQ and oxABTS after the addition of the HQ. Green-colored oxABTS changed back into colorless ABTS, which was accompanied by a decrease in the absorbance at 416 nm. At the same time, HQ was oxidized to the p-benzoquinone tautomer (PBQ) (Figure 3b).

To investigate the selectivity of the method, other substances, including cations, anions, and some phenols, were introduced into the system using the same experimental procedure. Compared with the signal of 30 μM HQ, the decrease in absorbance at 416 nm after the addition of most substances was not observed, demonstrating desirable selectivity for HQ detection (Appendix A). Similar to many earlier studies, catechol significantly interfered with hydroquinone detection [51,52,53]. The results indicate the developed method can analyze total hydroquinone and catechol in the presence of catechol. The practical application of the HQ sensor based on the MnO/PC nanohybrid was also demonstrated by adding a standard recovery experiment in the real sample. Appendix A shows that the recovery of the tested samples ranged from 96.0 to 106.3% and the RSDs were all below 4.6%. Thus, it is reliable to use the MnO/PC-based HQ sensor to detect HQ in practical applications.

## 4. Conclusions

Herein, an effective strategy for designing fungus-based laccase-mimicking nanozyme was proposed. A shell-like structured, MnO-loaded, porous carbon nanohybrid was synthesized through a simple calcination method. The MnO/PC hybrid catalyzed the oxidation of the laccase substrate ABTS and possessed excellent intrinsic laccase-like activities. Furthermore, the hybrid nanozyme played a crucial role in the ORR electrocatalytic activity with half-wave potentials of 0.73 V vs. RHE in 0.1 M KOH, limiting current density of −5.87 mA cm^−2^ and the electron transfer number to 4. Moreover, the colorimetric HQ sensor due to the oxidase-like activity of the hybrid nanozyme was developed, with the well linear relationship ranging from 0 to 50 μM and a detection limit of 0.5 μM. Our work provides a new and sustainable approach for the development of high-performance carbon–metal hybrid nanozymes.

## Data Availability

Data are available upon reasonable request to the authors.

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
