# Peer review of "Fungus-Based MnO/Porous Carbon Nanohybrid as Efficient Laccase Mimic for Oxygen Reduction Catalysis and Hydroquinone Detection"

_nanomaterials, 2022, doi:10.3390/nano12091596_

Round 1

Reviewer 1 Report

The manuscript entitled “Fungus Based MnO/Porous Carbon Nanohybrid as Efficient Laccase Mimic for Oxygen Reduction Catalysis and Hydroquinone Detection” discusses the synthesis and characterization of MnO /porous carbon nanohybrid. The manuscript can be published in Nanomaterials after revision. Please find my comments below.

1.    Experimental section. Line 74. “Tremella was purchased from local supermarket” is there any procedure of standardization of the material? How reproducible the results were?
2.    Line 84. Again “The tremella was purchased from the local supermarket” is repeated.
3.    Figure 1 closes the text in line 125.
4.     Section “Laccase-mimicking activity”. Please, compare the results with other, probably, standard material and give references.
5.    I think that Section “Sensing HQ based on MnO/PC nanohybrid” requires more detailed analysis and references.

Reviewer 2 Report

The author reported the synthesis of fungus-based MnO/carbon nanohybrid for ORR catalysis and hydroquinone detection. This is an interesting work. However, some statements need to be clarified and corrected. Thus, the manuscript should be improved by addressing the following comments.

  1. The authors prepared and characterized only MnO/PC hybrid without comparing single materials (i.e., MnO and PC). To confirm the merit and synergistic effects of MnO/PC on ORR and HQ detection, the authors should perform the characterization, electrochemical ORR measurement, and HQ detection of pure PC and MnO for comparison.
  2. Text (Page 4, Line 125) cannot be seen.
  3. EDS mapping analysis is recommended to confirm the presence of shell-like structure of MnO on the carbon surface (Figure 1B), 
  4. The diffraction peak of the XRD pattern in Figure 1D should be indexed.
  5. The ORR activity of MnO/PC should be compared with the MnO/PC (J. Phys. Chem. C, 2007, 111, 1434–1443; Chem. Euro J., 2019, 25, 2868–2876; Rep. 2015, 5, 8012, etc.) and laccase (i.e., Electroanal. 2004, 16, 1182–1185; Inorg. Chem., 2014, 53, 8505–8516, etc.) reported in the literature.
  6. The surface area and pore characters of the materials are known to play an important role in influencing the ORR activity. Therefore, the surface area and porosity analyses of MnO, PC, and MnO/PC are recommended.
  7. Please provide more in-depth discussion and information on the catalytic active sites for ORR on MnO/PC.
  8. What is the mechanism for detecting HQ on MnO/PC hybrid?
  9. Why did the ORR activity of MnO/PC-700 become lower? Please explain and discuss.
  10. What is the reason and purpose for annealing MnO/PC at 700 C?
  11. Figure 4A shows the absorption spectra in the presence of HQ concentration of 0–60 uM. However, the absorbance was plotted against the HQ concentration from 0 to 50 uM. Why was the HQ concentration of 60 uM not included?
  12. Peroxide species generated during ORR in alkaline solution Is HO2-, NOT H2O2. Therefore, it should be revised from H2O2 to HO2-.
  13. There are some typos and grammatical errors. Please carefully read and check.

Reviewer 3 Report

In this work, the author reported an effective strategy for designing fungus based laccase-mimicking nanozyme. A Shell-like structured MnO loaded porous carbon nanohybrid is reported which can catalyze the oxidation of the laccase substrate ABTS and possess excellent intrinsic laccase-like activities. MnO/PC also plays a crucial role in the ORR electrocatalytic activity with half-wave potentials of 0.73 V vs. RHE in 0.1 M KOH, limiting current density of -5.87 mA cm-2 , and the electron transfer number of 4. MnO/PC also showed colorimetric HQ sensor potential. 

This work provides a new and sustainable approach for the development of high-performance carbon-metal hybrid nanozymes. This is well arranged and discussed the own findings. 

I miss the comparative study in this work. The author should compare the physical and chemical properties of the MnO/PC with related materials. The author reported it as porous carbon with our surface properties. The potential of this material over other like-materials for oxidation of the laccase, ORR  and HQ sensing should be compared and discussed. The discussion should include the mechanistic study.

Thank you

Round 2

Reviewer 2 Report

I agree to accept this manuscript for publication in Nanomaterials.

Reviewer 3 Report

In this revised article, the author provided necessary information. I am convinced with the revised article. I am happy to recommend this for publication.

Thank you